The impact of three thioxothiazolidin compounds on trehalase activity and development of Spodoptera frugiperda larvae

Wu Yan 1
Hu Shangrong 2
Mao Qixuan 2
Shi Dongmei 3
Liu Xiangyu 1 2
Liu Busheng 2
Hua Liyuhan 2
Hu Gao 4
Li Can 1
Duan Hongxia hxduan@cau.edu.cn 3
Tang Bin tbzm611@hznu.edu.cn 2
1 Key Laboratory of Surveillance and Management of Invasive Alien Species, Guizhou Education Department, Department of Biology and Engineering of Environment, Guiyang University , Guiyang , China
2 College of Life and Environmental Sciences, Hangzhou Normal University , Hangzhou , China
3 Innovation Center of Pesticide Research, Department of Applied Chemistry, College of Science, China Agricultural University , Beijing , China
4 College of Plant Protection, Nanjing Agricultural University , Nanjing , China
Żyła Dagmara
Electronic publication date: 2024 Oct 9
Publication date: 2024
Volume: 12
Electronic Location ID: e18233
Received 2024 May 31; Accepted 2024 Sep 13
Copyright: ©2024 Wu et al.
Copyright year: 2024
Copyright holder: Wu et al.
License: This is an open access article distributed under the terms of the Creative Commons Attribution License, which permits unrestricted use, distribution, reproduction and adaptation in any medium and for any purpose provided that it is properly attributed. For attribution, the original author(s), title, publication source (PeerJ) and either DOI or URL of the article must be cited.
License URL: https://creativecommons.org/licenses/by/4.0/

Keywords: Spodoptera frugiperda, Trehalase inhibitor, Chitinase, Growth and development, Abnormal phenotype

Funding: National Key Research and Development Program of China 2023YFC2605200 Guiyang Science and Technology Personnel Training Project [2022]43-16 Program for Natural Science Research in Guizhou Education Department QJJ[2023]024 Guiyang Training Project for “Thousand Level” Talents in Guizhou Province GCC[2022]005 This work was supported by the National Key Research and Development Program of China (Grant No. 2023YFC2605200), the Guiyang Science and Technology Personnel Training Project [2022]43-16, the Program for Natural Science Research in Guizhou Education Department QJJ[2023]024, Guiyang Training Project for “Thousand Level” Talents in Guizhou Province GCC[2022]005. The funders had no role in study design, data collection and analysis, decision to publish, or preparation of the manuscript.

==============================
Trehalases (TREs), serving as crucial enzymes regulating trehalose and chitin metabolism in insects, represent prime targets for pest control strategies. We investigated the impact of three thioxothiazolidin compounds (1G, 2G, and 11G) on TRE activity and summarized their effects on the growth and development of Spodoptera frugiperda (Lepidoptera, Noctuidae). The experimental larvae of S. frugiperda were injected with the three thioxothiazolidin compounds (1G, 2G, and 11G), while the control group received an equivalent volume of 2% DMSO as a control. All three compounds had a strong effect on inhibiting TRE activity, significantly prolonging the pre-pupal development stage. However, compared with the 11G-treated group, the survival rate of larvae treated with 1G and 2G was significantly reduced by 31.11% and 27.78% respectively, while the occurrence of phenotypic abnormalities related to growth and development was higher. These results manifest that only the TRE inhibitors, 1G and 2G, modulate trehalose and chitin metabolism pathways of larvae, ultimately resulting in the failure molting and reduction of survival rates. Consequently, the thioxothiazolidin compounds, 1G and 2G, hold potential as environmentally friendly insecticides.

Introduction

Chitin, found in the extracellular matrix of insects and arthropods, plays a crucial role in insect molting, growth and development, as well as immunity (Muthukrishnan et al., 2019; Bouchebti et al., 2023). When the dynamic balance between chitin synthesis and degradation is disrupted or the chitin content changes in insects, abnormal molting or even death will occur (Zhang et al., 2017). The synthesis of chitin in insects begins with trehalase (TRE) and ends with chitin synthase (CHS) (Zhu et al., 2016; Chen et al., 2023). TRE is the only nonreductase responsible for irreversible degradation of trehalose, hydrolyzating 1 molecule of trehalose to two molecules of glucose, which is used as a substrate to participate in chitin synthesis (Neyman et al., 2022). Chitin biosynthesis is a complex process that requires TRE, Hexokinase (HK), Glucose-6-phosphate isomerase (G6PI), Glutamine: fructose-6-phosphate aminotransferase (GFAT), Glucosamine-6-phosphate N-acetyltransferase (GNPNA), 6-phosphate acetylglucosamine mutase (PAGM), UDP-N-acetylglucosamine pyrophosphorylase (UAP) and CHS work together (Liu, Zhang & Zhu, 2019). Therefore, TRE indirectly affect chitin synthesis by regulating the degradation of trehalose and affecting the levels of trehalose in insects. In insects, two distinct TREs have been characterized so far: soluble trehalase (TRE1) and membrane-bound trehalase (TRE2) (Shukla et al., 2015; Xu et al., 2022), which cooperate with each other to maintain energy supply in insects. Upon inactivation, chitinases (CHT), which catalyze the degradation of chitin in insects, can lead to serious defects in insect exoskeletons and affect insect development and growth (Chen & Yang, 2020). At present, TRE and chitinase have gradually become promising insecticide targets, playing an important role in the control of fungal pathogens and pest management (Oyeleye & Normi, 2018). Studies have confirmed that trehalose inhibitors can competitively bind to the active site of trehalose, forming complexes that inhibit the activity of trehalose, to inhibit the breakdown metabolism of trehalose in insects and achieve insecticidal effects (Tatun et al., 2014; Zhang et al., 2022). Validamycin A (García & Argüelles, 2021), trehazolin (El Nemr & ElAshryel, 2011; Mayack et al., 2020), some natural alkaloids and their analogues have been reported to effectively inhibit TRE activity in insects (Li et al., 2019a). While allosamidin (Viswanath et al., 2021), natural product argifin (Zhao et al., 2022) kasugamycin (Lee et al., 2022) have chitinase inhibitor effect, which provides new ideas for the development of green and environmentally friendly biological pesticides.

The fall armyworm (FAW), Spodoptera frugiperda (Lepidoptera: Noctuidae), a major agricultural pest of maize in North and South America, has gradually invaded the Eastern Hemisphere since the mid-1960s (Goergen et al., 2016; Wang et al., 2022a; Tay et al., 2023). The S. frugiperda has been classified as a significant migratory agricultural pest due to its wide host range, strong reproduction and migration ability, which usually cause indeliable damage to the invaded areas (Cui et al., 2019; Liu et al., 2022a). Chemical control has been an important measure for the prevention of S. frugiperda for decades (Wang et al., 2023). However, the frequent use of traditional pesticides, like various organophosphates, pyrethroids and carbamates, not only boosts the pest’s resistance to pesticides, but also adversely affects beneficial arthropods in the ecosystem. (Desneux, Decourtye & Delpuech, 2007; Jia et al., 2022; Haddi et al., 2023; Chen et al., 2024; Cai et al., 2024). Augmentative biological control using parasitoids and predators can mitigate pesticide hazards, providing green sustainable control of multiple major agricultural pests (Li et al., 2023a). The most common parasitoid of S. frugiperda is Trichogramma, and some predators, observed feeding on S. frugiperda eggs and larvae, such as spiders (Araneae), a tiger beetle (Coleoptera: Cicindelidae) and the pentatomid bug, Eocanthecona furcellata Wolf, can also help to control S. frugiperda infestation (Firake & Behere, 2020; Hou et al., 2022; Kenis et al., 2023; Nurkomar et al., 2024). The presence of insect entomopathogenic fungi, bacteria, nematodes and baculoviruses can interact with the insect immune system for promoting the natural protection of S. frugiperda and optimizing biological pesticide efficacy (Kenis et al., 2023; Zhang et al., 2024). Biological pesticides offer a more environmentally friendly alternative but still have limitations. For example, pyrethrum is an effective botanical insecticide with low toxicity to mammals. However, its high susceptibility to UV light poses a challenge, resulting in rapid loss of effectiveness (Harte et al., 2024). MicroRNAs (miRNAs) could inhibit the expression of CYP9F1 by binding to the 3′  untranslated region (3′  UTR) of CYP9F1 in S. frugiperda, improving the chlorotraniliprole susceptibility of S. frugiperda (Jiang et al., 2023a). Even insecticides, developed using plant secondary metabolites, could control the invasion and reproduction of pests by antifeeding responses of S. frugiperda (Pavela et al., 2023).

Inhibitors targeting trehalase have been shown to be effective in inhibiting the growth and development of S. frugiperda, even causing malformation or death. Validamycin (1 µg/µL), injected in S. frugiperda with microinjection, could significantly reduce trehalase activity 48 h later, resulting in reduction of the glucose and glycogen content (Luo et al., 2022). The compound 5K, a novel piperine derivative, has been shown to have substantial sublethal effects against Ostrinia furnacalis by regulating its growth and development (Jiang et al., 2022). Previous studies have confirmed that piperine derivates, such as ZK-PI-5 (C20H19NO4) and ZK-PI-9 (C19H16ClNO3), could act as trehalose inhibitors to regulate insect development and reproduction (Han et al., 2021; Wang et al., 2022b; Zhong et al., 2023). Three thiazolidinones with piperine skeletons (6a, 7b, and 7e), synthesized by introducing a thiothiazolidone structural fragment into the original piperine skeleton, has significant impact on the growth and development of S. frugiperda by prolonging the pupal stage (Tang et al., 2024). Therefore, piperine compounds were investigated with modifications as a potential inhibitor framework for TRE.

In this study, we substituted the thiothiazolidone moiety in compounds (6a, 7b, and 7e) with an acetic acid group and respectively combined methyl, ethyl, and propyl groups onto the carboxyl group of acetic acid moiety to obtain the thioxothiazolidin compounds 1G, 2G, and 11G. To assess the potential of compounds 1G, 2G, and 11G as TRE inhibitors, we investigated their effects on the growth and development of S. frugiperda larvae, aiming to provide novel insights for the development of environmentally friendly and sustainable biopesticides.

Materials & Methods

Insects

The S. frugiperda, offered by the Zhejiang Academy of Agricultural Sciences (Zhejiang, Hangzhou), was subsequently reared by our team in the Key Laboratory of Animal Adaptation and Evolution, School of Life and Environmental Sciences, Hangzhou Normal University (Zhejiang, Hangzhou). The specific rearing conditions and methods were consistent with those of Pinto et al. (2019); Tang et al. (2024).

Thioxothiazolidin compounds

The experimental TRE inhibitors 1G, 2G, and 11G were provided by the PMDD laboratory of China Agricultural University (Beijing). To achieve a concentration of 2 × 10−3 mmol/mL, dissolve these inhibitors in 2% DMSO , referring to the minimum effective concentration of TRE inhibitors on S. frugiperda measured by Wang et al. (2022b); Tang et al. (2024). Detailed information on the molecular weights and formulas of the TRE inhibitor can be found in Table 1, and their structural formulas are depicted in Fig. 1.

Table 1 Identification codes, molecular weight (MW), and molecular formulas of trehalase inhibitors.

Code	Purity	Solvent	MW	Molecular formula	
1G	98%	DMSO	337.364	C14H11NO5S2	
2G	98%	DMSO	357.442	C15H13NO5S2	
11G	98%	DMSO	377.857	C16H15ClNO5S2	

Figure 1 Molecular structures of three trehalase inhibitors: 1G, 2G, and 11G.

Microinjection of trehalase inhibitors

Using Sutter Instrument (Novato, CA, USA) to convert glass capillaries into microinjection glass needles, and subsequently installed these glass needles into the injection joint of the Eppendorf TransferMan®4r microinjection system and securely attached to the robotic arm. The S. frugiperda larvae were paralyzed on ice on the first day of the third instar. They were later then transferred to the loading platform of the microinjection system with a pointed brush and inserted 300 nL of the potential inhibitors into the thinner area between the second and third pairs of thoracic feet with needles compared to the control group treated with the equal volume of DMSO (2%). Treated larvae were reared in artificial climate boxes (Junyuan Experimental equipment store, Nantong, China).

Determination of trehalase activities

Samples were collected 48 h later for subsequent analyses after injecting 1G, 2G, 11G, and 2% DMSO. Approximately 15 larvae were collected randomly from each group and then mixed with 200 µL of phosphate-buffered saline (PBS; pH 7.0) for 30 min of sonication. The samples added 800 µL PBS were centrifuged at 1,000 × g at 4 °C for 20 min, followed by collection of approximately 350 µL of supernatant for another 60 min ultracentcentrifugation. The activity of soluble trehalase was determined with another 300 µL of the resulting supernatant, and the rest precipitate was mixed with 300 µL of PBS for subsequent determination about the activity of membrane-bound trehalase, following the Gglucose (Go) Assay Kit instructions (Lot No. SLCD8160, Sigma, St. Louis, MO, USA). Three biological replicates were performed for each group.

Determination of chitin content

Collecting larvae in Eppendorf (EP) tubes 48 h after treatment. Three biological replicates and three technical replicates were contained in each experiment. The samples, added with 500 µL of 6% KOH solution, were incubated at 80 °C for 90 min and then centrifuged at 12,000 × g for 20 min at 4 °C after 5 min of shaking. The resulting precipitate was resuspended in 200 µL McIlvaine’s buffer (Macklin, China). The chitin was hydrolyzed by adding 10 µL of chitinase from Streptomyces griseus (Sigma-Aldrich, USA), and the mixture was incubated at 37 °C and 150 rpm for 72 h in a shaker.

After completion of the hydrolysis reaction, the samples were centrifuged at 12,000 × g for 1 min at 25 ° C. The collecting supernatant (60 µL) was added with an equal volume of sodium borate. Stirring and incubating the mixture in a water bath (99.9 °C) for 10 min, followed by cooling to 25 °C. A total of 600 µL of 1 × DMAB was added into the mixture for later incubation in a water bath at 37 °C for 20 min. Sample (200 µL) were added to the wells of enzyme-linked plates for determination of the absorbance at 585 nm. The standard curve was drawn by measuring the absorbance value of the standard solution for calculating the chitin concentration. This procedure was referred to the determination of chitin content by Yao et al. (2010).

Determination of chitinase activities

Collecting larvae in EP tubes after 48 h of treatment, with three biological replicates and three technical replicates for each experiment. The tissue (approximately 0.1 g) was homogenized in one mL of ice-cold extraction solution for 20 min centrifugation at 10,000 × g at 4 °C. The 400 µL of supernatant was collected on ice for chitinase activity testing with a chitinase reagent kit (Suzhou Keming Biotechnology Co., Ltd). Chitinase activity (mg/h/g fresh weight) was calculated according to the formula of Tang et al. (2024): Chitinase activity=ΔA+0.2753÷6.4108×Vtotal reaction system÷Vsample÷Vtotal sample×W÷T×10=3.899×ΔA+0.2753÷W

(A, absorbance; ΔA =Ameasured value - Acontrol value; W, sample weight).

Observation of the feeding, growth, and development of treated S. frugiperda larvae

The initial treatment numbers for the 2% DMSO, 1G, 2G, and 11G groups were all 90. Larvae were selected and fed separately every 2 days with an artificial diet in a feeding box on the first day of the third instar. The larval shells were used for determination of the larval age and dislodged after data recording to avoid confusion. Every 24 h, the weight, emergence, and mortality rate of the larvae were recorded on a daily basis and record the remaining number of larvae for each treatment on a record sheet. Calculate the survival rate based on the recorded data and observe the phenotype of larvae after pupation.The formula for calculating survival rate is as follows: Survival rate at different ages=number of survivors at that age÷number of initial treatments×100%.

Larvae were observed daily to record their growth and development since the first day after injecting TRE inhibitors. The weights of larvae were measured, as well as the survival, deformity, pupation, and emergence rates were recorded when pupae formed. The images of abnormal larvae were taken with a Canon EOS 50D to record phenotypic changes, and categorized into lethal and non-lethal phenotypes based on the lethality of the observed abnormalities.

Quantification and statistical analyses

The IBM SPSS Statistics 20 software (SPSS v. 20, Armonk, NY, USA) was used to perform statistical analyses. The differences between control and treatment groups were compared by one-way analysis of variance (ANOVA) or Student’s t-test. Post-hoc tests were performed with the Tukey. Statistical significance was set at p < 0.05. The data were expressed as mean ± standard error. After data analysis, GraphPad Prism software, version 8.0, was used to generate graphs. Data analysis methods were based on those previously described by Tang et al. (2024).

Results

Thioxothiazolidin compounds inhibit trehalase activity in S. frugiperda larvae

Thioxothiazolidin compounds (1G, 2G, and 11G) were injected into the third instar larvae of S. frugiperda at a concentration of 2 × 10−3 mmol/mL, and samples were analyzed for the detection of TRE activity at 24 h. The results showed that 1G, 2G, and 11G inhibited TRE2 activity compared to the control group, but had no significant effect on TRE1 (Figs. 2A, 2C). After a 48-hour injection period, both TRE1 and TRE2 activities were downregulated in the larvae treated with three inhibitors (Figs. 2B, 2D). Of the tested compounds, 1G and 2G displayed the most pronounced inhibitory effects on TRE2 activity, suggesting their potential as strong inhibitors (Fig. 2D).

Figure 2 Effects of trehalase (TRE) inhibitors on TRE activity in S.  frugiperda larvae 24 h and 48 h after treatment.

Changes in soluble TRE activities after (A) 24 h and (B) 48 h injection. Changes in membrane-bound TRE activities after (C) 24 h and (D) 48 h injection. Values are presented as the means ± SE. ∗p < 0.05 and ∗∗p < 0.01 denotes significant differences. ns: not significant (independent samples; t-test); 2% DMSO: negative control group; 1G, 2G, and 11G: treatment groups.

Thioxothiazolidin compounds affect chitin content and chitinase activity in S. frugiperda larvae

The chitin content in the 11G group was significantly lower than that in the control group, whereas there was no significant difference between the control and 1G and 2G groups after 24 h injection (Fig. 3A). Compounds 2G and 11G significantly reduced the chitin content of S. frugiperda larvae injecting 48 h later, whereas 1G had no significant effect (Fig. 3B). The activity of chitinase was significantly upregulated in the 1G group treated 24 h later. In contrast, at the 48-hour injection point, the chitinase activity of 1G and 11G groups decreased sharply, while that of 2G group increased significantly compared with the control group (Figs. 3C, 3D).

Figure 3 Effects of trehalase (TRE) inhibitors on chitin content, chitinase activity in S.  frugiperda after 48 h of treatment.

Chitin content after (A) 24 h and (B) 48 h injection. Chitinase activity after (C) 24 h and (D) 48 h injection. Values are presented as the means ± SE. ∗p < 0.05 and ∗∗p < 0.01 denote significant differences. ns, not significant (independent samples; t-test); 2% DMSO: negative control group; 1G, 2G, and 11G: treatment groups

Thioxothiazolidin compounds affect the development of S. frugiperda larvae

No significant differences were observed in the developmental time of the fourth, fifth, sixth instar larvae, as well as the pupal stage. Nevertheless, within the experimental cohorts, the pre-pupal larval stage exhibited a marked extension in development time specifically within the 1G, 2G, and 11G groups. Throughout the transformation of S. frugiperda larvae from the third instar to the adult stage, there was no statistically significant disparity in the cumulative growth and development period when comparing larvae subjected to inhibitors with the control group. However, the total developmental duration was significantly different between 1G and 2G treatments (Table 2).

Table 2 The developmental durations at each instar in S. frugiperda after injection of inhibitors.

Treatment	4th instar	5th instar	6th instar	Pre-pupation	Pupation	Overall developmental duration	
2% DMSO	2.45 ± 0.22a	2.93 ± 0.01a	4.14 ± 0.15a	1.12 ± 0.13a	9.37 ± 0.07a	19.79 ± 0.18ab	
1G	2.22 ± 0.13a	2.75 ± 0.33a	4.17 ± 0.50a	1.85 ± 0.20b	9.93 ± 0.28a	20.74 ± 0.81b	
2G	2.24 ± 0.12a	2.66 ± 0.16a	3.26 ± 0.30a	1.84 ± 0.28b	9.57 ± 0.49a	19.50 ± 0.05a	
11G	2.45 ± 0.18a	2.61 ± 0.29a	3.63 ± 0.41a	1.42 ± 0.26ab	9.87 ± 0.09a	19.86 ± 0.35ab	
Notes.

Durations are presented in days. Superscript letters denote significant differences (p < 0.05) from the control group. The analysis included comparisons among the four different treatments within the same S. frugiperda larvae. Values are presented as means ± SE (ANOVA followed by Tukey’s post hoc test). 2% DMSO: negative control group; 1G, 2G, and 11G: treatment groups.

Larval weight changes were assessed from the fourth instar to the pupal stage in S. frugiperda. The figure indicated that larval weight in the fourth instar was observably different between 1G and 11G treatments, but none of the inhibitors altered the larval weight compared with the control (Fig. 4A). On the contrary, there was a significant increase in larval weight at the sixth stage in the inhibitor treatment groups (Fig. 4C). The weight of the 2G and 11G groups was significantly higher than that of the other two groups in the fifth instar (Fig. 4B). Larval weight at the pre-pupal and pupal stage, treated with 1G and 2G, was significantly higher than in the 11G and control groups (Figs. 4D, 4E).

Figure 4 Effect of trehalase (TRE) inhibitors on the weight of each instar in S. frugiperda.

(A) Fourth, (B) fifth, and (C) sixth instar. (D) Pre-pupa and (E) pupa stages. Values are presented as the means ±  SE. ∗p < 0.05 denotes significant differences. ns, not significant (independent samples; t-test); 2% DMSO: negative control group; 1G, 2G, and 11G: treatment groups.

Thioxothiazolidin compounds decrease the survival rate of S. frugiperda larvae

The survival rate underwent a decline from the third instar to the pre-pupal stage, culminating in a 67.78% survival rate in the control group. Subsequently, a notable decrease in survival rate was observed from the pre-pupal stage to adulthood, yielding a 54.44% survival rate among adult moths (Fig. 5A).The death of larvae in the 2G group mainly happened between the sixth instar and emergency stage, resulting in a survival rate of 32.22%, less than half compared to that at initial treatment (Fig. 5A). It is noteworthy that the larvae belonging to the 1G group demonstrated significantly high mortality rates throughout their developmental stages, spanning from the third instar to the pupal stage. Consequently, at the pupal stage, the survival rate of these larvae amounted to a mere 34.44%. Furthermore, among the larvae in the 1G group that successfully progressed into adult moths, they constituted a mere 28.89% of the initially treated larvae, which is a notably lower proportion compared to that observed in the 2G, 11G, and control groups. (Fig. 5A). However, the final fatality rate of the 11G group reached 60%, higher than that of the control group, indicating there was no remarkable effect on larvae mortality in the 11G group (Fig. 5A). The pupation rate exhibited significant variations in the 1G group compared to the other groups, with the exception of the 2G group. Conversely, no discernible differences were observed among the remaining groups (Fig. 5B). Despite the increased larval mortality observed in the 1G and 2G groups, no statistically significant differences were detected among the groups in terms of emergence rates (Fig. 5C).

Figure 5 Effect of trehalase (TRE) inhibitors on the survival, pupation, and emergence rates of each instar in S.frugiperda.

(A) Survival, (B) pupation, and (C) emergence rates. Values are presented as the means ±  SE. Pupation rate = (pupal number/sixth instar larvae number) ×100%; Emergence rate = (number of adults/pupal number) ×100%. Values are presented as the means ± SE. ns, not significant (independent samples; t-test); 2% DMSO: negative control group; 1G, 2G, and 11G: treatment groups.

Thioxothiazolidin compounds increase the rate of abnormal phenotypes and deformity in S. frugiperda larvae

In all treatment groups, from the fourth instar stage to the pupal stage, observations were made revealing abnormal phenotypes of larvae (Fig. 6). Furthermore, the aberration rates were notably elevated in comparison to those observed in the control group (Table 3). Abnormal phenotypes, compared to normally developing larvae, occurred during the development in groups 1G, 2G, and 11G, resulting in what was termed an abnormal non-lethal phenotype (Fig. 6A). In particular, larvae in the 1G group exhibited a ruptured outer epidermis with exposed contents at the fourth instar, accompanied by a darkening abdomen at the pre-pupal period. A similar condition, manifesting as distinct invagination or folding of the outer epidermis, was also observed in the fourth instar larvae of group 2G (Fig. 6B). This phenotype of death due to molting failure was considered as an abnormal lethal phenotype.

Figure 6 Effect of trehalase (TRE) inhibitors on the phenotype of S.  frugiperda larvae.

(A) Abnormal non-lethal phenotype, and (B) abnormal lethal phenotype. The same larvae were tracked to observe abnormal non-lethal phenotypes across different treatments. The number of deaths during development were counted and all abnormal death phenotypes and abnormal non-death phenotype were recorded. Values are presented as the means ± SE (independent samples; t-test). 1G, 2G, and 11G: treatment groups.

Table 3 Abnormality rates at each instar of S. frugiperda after injection of three trehalase inhibitors.

Treatment	Number of larvae (n)	Abnormality rate (%)	p -values	
2% DMSO	88	3.33	–	
1G	90	16.67	p= 0.003b	
2G	89	11.11	p= 0.044b	
11G	90	14.44	p= 0.009b	
Notes.

Comparisons were made between the four different treatments using the same S. frugiperda larvae. Superscript denote significant differences (p < 0.05) from the control group (chi-square test). Values are presented means ±  SE (ANOVA followed by Tukey’s post hoc test). 2% DMSO: negative control group; 1G, 2G, and 11G: treatment groups.

Discussion

Trehalose in insects has important biological functions. Its metabolic process of trehalose can not only regulate the chitin biosynthesis to control the growth and development in insects, but also store energy and assist in the recovery of adversity (Tang et al., 2017; Tang et al., 2018; Shi et al., 2022). Trehalase, the only important functional enzyme that hydrolyzes trehalose, is essential for the regulation of trehalose content (Sakaguchi, 2020). The main active ingredients of traditional chemical insecticides and some biological pesticides include lambda cyhalothrin, spinotram, emamectin tebufenozide, Bacillus thuringiensis and other chemical components, their extensive use can have negative impacts on human health and the environment (Hou et al., 2024). New insecticides/fungicides, which are safe and effective against plants and mammals, can be developed by targeting TRE. Several TRE inhibitors have been extensively studied as potential insecticides. For example, validamycin could inhibit trehalase activity, leading to the suppression of fungal growth and insecticidal effects (Li et al., 2019b; Yang et al., 2023). Trehazolin was injected into the haemolymph of locusts, irreversibly inhibiting TRE activity in locust flight muscle (Wegener et al., 2010). A large number of studies have also confirmed that validamycin A, like validamycin, exhibit similar inhibitory effects on Lepidoptera, Diptera, Hemiptera, and other insects (Tang et al., 2017; Marten et al., 2020; García & Argüelles, 2021; Li et al., 2023b).

In this research, we treated larvae with three compounds 1G, 2G, and 11G and found that all three inhibitors competitively bind to the TRE in S. frugiperda, contributed to the inhibition of TRE activity in larvae after 48 h of treatment (Figs. 2B, 2D). This inhibitory effect was not so evident 24 h after injection in TRE1, while only TRE2 activity was significantly inhibited (Figs. 2A, 2C), demonstrating that the inhibitors might have a stronger affinity for TRE2. These results certified the potential of thiazolidinone as TRE inhibitors (Wang et al., 2022b; Zhong et al., 2023; Jiang et al., 2023b).

Chitin serves as the major component of the cuticle and acts as a physical barrier against predation in insects, which content is related to chitin synthase and chitinase (Noh et al., 2016; Veliz, Martínez-Hidalgo & Hirsch, 2017; Muthukrishnan et al., 2020). The TRE inhibitors, 1G and 11G, had not significantly inhibited chitinase activity compared to the control group until 48 h had elapsed, and the content of chitin in the corresponding treatment groups was slightly higher than that of 24 h (Fig. 3). These findings confirm the important roles of chitinase in the regulatory mechanism of chitin biosynthesis (Noh et al., 2016; Veliz, Martínez-Hidalgo & Hirsch, 2017). Previous studies have shown that insect chitinases can be divided into 11 groups, belonging to glycoside hydrolase family 18 (GH18)3 (Tetreau et al., 2015). Barbole et al. (2024) found that berberine-fed S. frugiperda exhibit higher expression of SfCht5 and chitinase 7, and both silencing and overexpression of SfCht5 lead to dysregulation of the ecdysone receptor, resulting in defects in S. frugiperda development. The chitinase-h, ofChi-h, in Ostrinia furnacalis was found inhibited by N,N′,N″-trimethylglucosamine-N,N′, N″, N″′-tetraacetylchitotetraose (TMG-(GlcNAc)4), leading to severe allergic reactions in the injection groups. Most abnormal pupae and larvae die within 10 days after injection (Liu et al., 2017). Some types of fungi produce cyclic pentapeptides called argifin and argadin, which inhibit insect chitinase at nM concentrations. These peptides can inhibit chitinase in fungi and humans, while also affecting the molting of cockroach larvae, indicating their potential as fungicides/insecticides (Rao et al., 2005; Arakane & Muthukrishnan, 2010). The three TRE inhibitors may severely disrupt the balance of chitin metabolism in insects, with 1G and 11G being the most effective in inhibiting chitinase (Fig. 3).

In addition, the growth, development, and reproduction of insects are influenced by various biological and abiotic factors (Hafsi & Delatte, 2023). The survival rate of newly emerged S. frugiperda larvae treated with 1G and 2G was less than 40% (Fig. 5A), primarily owing to abnormal chitin metabolism (Liu et al., 2022b). Coincidentally, an abnormal phenotype of epidermal rupture was found in fourth instar larvae following 1G and 2G injections (Fig. 6B). This abnormality contributed to a sharp decrease in the survival rates of the third to fourth instar (Fig. 5A). The findings were similar to the results of Zhong et al. (2023) in which the proportion of abnormal phenotypes characterized by emergence failure increased when the TRE inhibitors ZK-PI-5 and ZK-PI-9 were injected, possibly due to contraction of abdominal segments caused by the loss of the outer epidermis of the larvae. Camptothecin inhibited the weight gain of lepidopteran larvae and their development of Mythimna separata and S. frugiperda in a concentration-dependent manner (Ding et al., 2023). RNA interference (RNAi)-mediated silencing of HcCht5 can inhibit the molting of Hypantria cunea larvae after double-stranded RNA (dsRNA) injection (Zhang et al., 2021). Prothoracicotropic hormone (PTTH), produced by neurosecretory cells in the insect brain, plays an important role in insect growth and metamorphosis, and can be used as a potential target for pest control. The developmental time of larval has been significantly prolonged and even death during the larvae molting and pupation by knocking out PTTH with RNAi or CRISPR/Cas9 (Li et al., 2023c).

All three TRE inhibitors 1G, 2G, and 11G, had no significant effect on pupation or emergence rates (Figs. 5B, 5C), suggesting that the late larval stages were not significantly affected by the TRE inhibitors. Furthermore, S. frugiperda larvae treated with the TRE inhibitors exhibited non-lethal abnormal phenotypes ranging from the fourth instar to the pupal stage. However, there is currently no clear mechanism to explain this phenotypic difference. Therefore, we still need to invest significant time and effort in developing technologies to explore this mechanism.

Conclusions

In summary, TRE and chitinase serve as crucial functional enzymes in insects, profoundly affecting the survival and normal development of insects when their activities are inhibited. All three inhibitors have the potential to decrease the activity of TRE in S. frugiperda larvae, subsequently causing variable impacts on chitinase activity and, consequently, affecting larval development. The findings indicate that the thiothiazolidines compounds, 1G and 2G, play significant roles in the malformation and mortality of larvae, while the inhibitory effect of 11G was not obvious. Consequently, the compounds 1G and 2G emerge as promising candidates for environmentally friendly insecticides, offering significant potential for integration into sustainable pest control strategies.

Supplemental Information

Supplemental Information 1 Picture of phenotypes

Supplemental Information 2 Raw data for Figures

Additional Information and Declarations

Competing Interests

Author Contributions

Data Availability

The authors declare there are no competing interests.

Yan Wu conceived and designed the experiments, authored or reviewed drafts of the article, supervision, and approved the final draft.

Shangrong Hu conceived and designed the experiments, analyzed the data, prepared figures and/or tables, authored or reviewed drafts of the article, and approved the final draft.

Qixuan Mao performed the experiments, prepared figures and/or tables, visualization, and approved the final draft.

Dongmei Shi analyzed the data, prepared figures and/or tables, visualization, and approved the final draft.

Xiangyu Liu performed the experiments, prepared figures and/or tables, investigation, and approved the final draft.

Busheng Liu performed the experiments, prepared figures and/or tables, methodology, and approved the final draft.

Liyuhan Hua performed the experiments, prepared figures and/or tables, methodology, and approved the final draft.

Gao Hu performed the experiments, authored or reviewed drafts of the article, supervision, and approved the final draft.

Can Li conceived and designed the experiments, authored or reviewed drafts of the article, supervision, and approved the final draft.

Hongxia Duan conceived and designed the experiments, authored or reviewed drafts of the article, supervision, and approved the final draft.

Bin Tang conceived and designed the experiments, authored or reviewed drafts of the article, supervision, and approved the final draft.

The following information was supplied regarding data availability:

The raw measurements are available in the Supplemental Files.

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
