# Peer review of "The impact of three thioxothiazolidin compounds on trehalase activity and development of Spodoptera frugiperda larvae"

_PeerJ, doi:10.7717/peerj.18233_

## Round 0.1 · original submission · Major Revisions

· Academic Editor

Major Revisions

The manuscript requires major revision because it is lacking some important methodological aspects. Please, address all the reviewers' comments and suggestions.

Reviewer 1 ·

Basic reporting

no comment

Experimental design

no comment

Validity of the findings

no comment

Additional comments

The manuscript is well written. There are some comments that may improve the manuscript as follow.
1. line 147; which larval instar that had been treated with the inhibitors and please specify the number of specimen.
2. line 153; change the word "trehalose" to "trehalase"
3. line 181: which larval instar that was used in this experiment?
4. line 277: change the word "trehalose" to "trehalase"
5. Fig 2, the unit of trehalase activity in A, B, C, and D should be the same please correct.
6. I read the article reported by Zhong et al. 2023, the activity of soluble trehalase in 3rd instar larvae was about 0.004 nmole glucose/ ug protein/ min, however in this manuscript the activity was detected about 0.01nmole glucose/ ug protein/ min. Please provide the reason for this difference.
7. Fig. 3, please check the unit of chitin, what per means?
8. Fig. 5, the color of each bar in Fig. 5B and 5C should correlated with the color used in Fig.5A
9. Please explain more how to calculate the survival rate of pupa in Fig. 5A.

·

Basic reporting

Reviewer’s comment
Comments to the Authors
Thanks for your manuscript [The impact of three thioxothiazolidin compounds on trehalase activity and development of Spodoptera frugiperda larvae], but it should be carefully modified because of the following points;
-Scientific names must be written correctly. The full scientific name of an insect (animal) species that is being studied must be given together with the authority and the order and family placement when first mentioned in the abstract and the main text. The genus name should be abbreviated thereafter with the exception that the full form of the scientific name should be used when beginning a sentence.
-Scientific names must be written in italic.
-Introduction section can be supported with some more recent related literature. The purpose of the study should be clearly stated at the end of the Introduction; Author should explain about the novelty of the work
-Materials and methods section; Give information about the geographical location of study area.
- Result section is required to explain the findings of the study more extensively.
- The article needs to re-update the references, especially in the last years, so that it does not lack contemporary.

Experimental design

'No comment'

Validity of the findings

'No comment'

Additional comments

Reviewer’s comment
Comments to the Authors
Thanks for your manuscript [The impact of three thioxothiazolidin compounds on trehalase activity and development of Spodoptera frugiperda larvae], but it should be carefully modified because of the following points;
-Scientific names must be written correctly. The full scientific name of an insect (animal) species that is being studied must be given together with the authority and the order and family placement when first mentioned in the abstract and the main text. The genus name should be abbreviated thereafter with the exception that the full form of the scientific name should be used when beginning a sentence.
-Scientific names must be written in italic.
-Introduction section can be supported with some more recent related literature. The purpose of the study should be clearly stated at the end of the Introduction; Author should explain about the novelty of the work
-Materials and methods section; Give information about the geographical location of study area.
- Result section is required to explain the findings of the study more extensively.
- The article needs to re-update the references, especially in the last years, so that it does not lack contemporary.

·

Basic reporting

The manuscript provides detailed information on how three thioxothiazolidin compounds affect the activity of the enzyme trehalase in relation to the larval development of Spodoptera frugiperda.
• The introduction briefly outlines the mode of action of trehalase concerning chitin synthase. This requires a more detailed elaboration.
• The author has discussed RNAi and miRNA in the introduction. How do these concepts relate to the current context of the study?
Line 114 mentions the development of environmentally friendly and sustainable biopesticides. How does the present work ensure this goal?

Experimental design

Line no. 161 mentions the source of chitinase from Streptomyces griseus. It is necessary to specify whether the enzyme was commercially obtained or to specify the exact source or isolate of Streptomyces griseus

Line no. 177 describes the quantification of chitinase enzyme. What does T stands for?

Line 273 discusses traditional insecticides, mentioning Bacillus thuringiensis as one of the components. The statement suggests that the extensive usage of Bacillus thuringiensis may have negative impacts on human health and the environment. Recheck the statement/citation of Hou et al., 2024

References No. 5, 7, 12, 15, 22, 23, 29, 35, 40, 43, 54, 56, 62, 64, 67, and 69 are not cited according to the journal's guidelines.

Validity of the findings

The observations recorded show a significant increase in larval development in the treated groups compared to the untreated control groups. However, there was a decrease in the survival rate, increase in the rate of abnormal phenotypes and no significant effect on pupation and emergence rates. These findings require thorough discussion with proper justifications.

---

## Round 0.2 · Minor Revisions

· Academic Editor

Minor Revisions

Please, accommodate the remaining minor comments of Reviewer 1.

Reviewer 1 ·

Basic reporting

-

Experimental design

-

Validity of the findings

-

Additional comments

Thank you for the revised version of manuscript. The authors incorporated most of the comments and the manuscript was improved obviously. These are my additional comments:
1. Please reconsider about the RNAi that should be removed from the manuscript because all experiments was not involve with the RNAi. If the author intend to mention about the effect of trehalase inhibitors treatment that is similar to the RNAi (trehalase/ chitinase genes) treatment which cause the decrease in trehalase activity and chitin metabolism, please rewrite and provide some references to support. In addition, if it's possible to combine the application of RNAi with these trehalase inhibitors in the future research to obtain more insecticidal activity, the authors can mention in the discussion section.
2. Please correct Fig.4 E, should it be pupal weight or larval weight?
3. It would be great if the authors additionally explain why the concentration of TRE inhibitors was 2x10-3 mmol/mL. The readers will understand easily if the inhibitory activity against varied concentrations of 1G, 2G and 11G was examined and noted anywhere in the manuscript.

·

Basic reporting

Dear Editor,
I hope you're doing well, the author did the requested revisions.
Best Regards,
Prof. Seham Mansour Ismail

Experimental design

No comment

Validity of the findings

No comment

Additional comments

No comment

---

## Round 0.3 · accepted · Accept

· Academic Editor

Accept

Thank you for addressing all the reviewers' comments. I am happy with the current version and in my opinion, the manuscript is ready for publication and does not require another round of review.